# Mycotoxins and the Enteric Nervous System

**DOI:** 10.3390/toxins12070461

**Published:** 2020-07-19

**Authors:** Sławomir Gonkowski, Magdalena Gajęcka, Krystyna Makowska

**Affiliations:** 1Department of Clinical Physiology, Faculty of Veterinary Medicine, University of Warmia and Mazury in Olsztyn, Oczapowskiego 13, 10-957 Olsztyn, Poland; slawomir.gonkowski@uwm.edu.pl; 2Department of Veterinary Prevention and Feed Hygiene, Faculty of Veterinary Medicine, University of Warmia and Mazury in Olsztyn, Oczapowskiego Str. 13, 10-718 Olsztyn, Poland; magdalena.gajecka@uwm.edu.pl; 3Department of Clinical Diagnostics, Faculty of Veterinary Medicine, University of Warmia and Mazury in Olsztyn, Oczapowskiego 14, 10-957 Olsztyn, Poland

**Keywords:** mycotoxins, enteric nervous system, gastrointestinal tract, mammals, animal pathology, intestines, toxins, feed

## Abstract

Mycotoxins are secondary metabolites produced by various fungal species. They are commonly found in a wide range of agricultural products. Mycotoxins contained in food enter living organisms and may have harmful effects on many internal organs and systems. The gastrointestinal tract, which first comes into contact with mycotoxins present in food, is particularly vulnerable to the harmful effects of these toxins. One of the lesser-known aspects of the impact of mycotoxins on the gastrointestinal tract is the influence of these substances on gastrointestinal innervation. Therefore, the present study is the first review of current knowledge concerning the influence of mycotoxins on the enteric nervous system, which plays an important role, not only in almost all regulatory processes within the gastrointestinal tract, but also in adaptive and protective reactions in response to pathological and toxic factors in food.

## 1. Introduction

Mycotoxins are a group of several biochemicals synthesized as secondary metabolites by various species of fungi [1]. They are commonly found in a wide range of agricultural products, such as cereals (maize, wheat, rye), fresh and dried fruits, grape juice, spices, herbs and many others [2,3,4]. Moreover, the presence of mycotoxins has also been observed in food products of animal origin and water [3,4,5,6]. Previous studies have shown that mycotoxins show multidirectional harmful effects on human and animal health. It is known that mycotoxins may act on many internal organs and systems, including, among others, nervous, reproductive and immunological systems, metabolic processes and endocrine glands [7].

This widespread occurrence of mycotoxins and their adverse effects demonstrate that these substances are a serious health and economic problem of the contemporary world and therefore, mycotoxins are the most widely studied biological toxins [5,6]. However, many aspects of mycotoxin activity on eukaryotic organisms are unknown. One lesser-known issue is the influence of these substances on the enteric nervous system (ENS).

Since mycotoxins are present in food and drinking water, the gastrointestinal (GI) tract is the part of the body that first comes into contact with these toxic factors [8]. A relatively large number of studies have described mycotoxin-induced morphological and functional changes in the GI tract, whose character depends on the type of mycotoxin, mammal species studied, as well as the degree and length of exposure to mycotoxins [9,10,11,12,13,14,15,16,17,18,19]. The most common effects of mycotoxins on the GI tract include inflammatory and necrotic changes, disturbances in secretory activity and metabolism of the enterocytes, damage to the intestinal barrier and dysfunction in intestinal absorption [10,11,16,20]. Unfortunately, the impact of mycotoxins on the ENS has been neglected in toxicological studies for many years. There are a few recent studies published which describe this aspect of mycotoxin activity. These reports have indicated that the ENS plays a crucial role in the regulation of the majority of gastrointestinal functions, takes part in adaptive and protective processes and is one of the first barriers against pathological and toxic factors in food [15,16,17,21,22] and may also be compromised by the harmful effects of mycotoxins. Therefore, this work is an attempt to summarize the influence of mycotoxins on the ENS. To better understand this influence, a short description of the organization of the ENS is needed.

## 2. Anatomy of the Enteric Nervous System

The enteric nervous system is a specific part of the autonomic nervous system. It is situated in the wall of the gastrointestinal tract from the esophagus to the rectum and is responsible for the majority of gastrointestinal activities [23]. In terms of the number of nerve cells, the ENS is the second largest (after the brain, and before the spinal cord) nervous structure in mammals, which may contain an estimated 200–500 million neurons [24,25,26]. For this reason, as well as due to the complicated structure and high autonomy, the ENS is often called the intestinal brain [24].

Millions of neurons comprising the ENS are grouped in the neuronal ganglia, which are interconnected with a dense network of nerve fibers and form ganglionated plexuses. The localization and number of these plexuses depend on the mammal species and the segment of the GI tract. In rodents, the ENS in the esophagus and stomach is built of two types of intramural ganglia. The first type, the myenteric ganglion, is located between longitudinal and circular muscle layers. Myenteric ganglia are interconnected with a dense network of nerve fibers and form the myenteric plexus [27,28,29,30]. The second type of intramural ganglia, the submucous ganglion, is located in the submucous layer, near the muscularis mucosae of the mucosal layer. Contrary to muscular ganglia, the nerves interconnected with the submucous ganglia are rather sparse. Therefore, submucous ganglia in the esophagus and stomach do not form plexus [31], although some authors have described submucous plexus in rodent esophagus and stomach [32,33]. However, the situation is different in the small and large intestines in rodents. Both types of enteric ganglia (myenteric and submucous) located in the same places as in the esophagus and stomach are interconnected with a dense network of nerves. Therefore, two kinds of plexuses (the myenteric plexus and submucous plexus) are described in the rodent intestine [34,35,36,37].

In large mammals, the organization of the ENS in the esophagus and stomach is similar to rodents [38,39,40], although some authors have described three types of the enteric plexuses (such as in the intestine—see below) in the porcine stomach [41]. The only exception are ruminants, in which only one type of the enteric ganglia (myenteric ganglia) has been described in the forestomach. These ganglia are located between longitudinal and circular muscular layers, interconnected with the dense nerve fibers and form myenteric plexus [42,43].

In turn, there are three types of the enteric ganglia, which form intramural plexuses in the small and large intestine of large mammal species (for example, in the pig) (Figure 1) [44,45,46]. The first of them is the myenteric plexus located (similarly to rodents) between the longitudinal and circular muscle layer [45,46]. Moreover, two types of submucous plexuses located in the submucous layer of the intestinal wall have been observed: outer submucous plexus—located in close association with the adjacent circular muscle layer (on its inner side) and the inner submucous plexus—positioned closer to the intestinal lumen, near the muscularis mucosae [47,48,49]. These plexuses are also often named after their discoverers. The myenteric plexus often called Auerbach’s plexus, the outer submucous plexus—Schabadash’s plexus, and the inner submucous plexus (in rodents—the submucous plexus)—Meissner’s plexus [50,51].

As regards the organization of the human ENS, the distribution of the nervous structures in the esophagus and stomach is similar to rodents and large mammal species [23,31,52,53]. In the human small and large intestines, the organization of the ENS is not quite clear. Previous publications have described four types of enteric plexuses. In addition to the above-mentioned myenteric, outer submucous and inner submucous plexuses, the presence of an intermediate submucous plexus (IMSP)—a ganglionated plexus located in the submucous layer between the outer and inner submucous plexus has been reported [54]. However, at present, three kinds of plexuses located similarly to the porcine intestine are described in the human small and large intestine. In addition to the myenteric plexus located between the longitudinal and circular muscle layers, they include the plexus submucosus externus (PSE) near the circular muscle layers (on its inner side) and plexus submucosus internus (PSI) located closer to the intestinal lumen [23,31,55,56,57,58,59]. Contrary to the porcine inner submucous plexus, PSI in the human intestine is multi-layered, which means the particular ganglia within this plexus are located at a different depth of the submucous layer [56]. Other publications have shown that submucosal ganglia in the human colon are disseminated throughout the submucosal layer with significant inter-individual differences [60].

In addition to the above-mentioned main types of enteric ganglia, previous studies conducted in various mammal species have also reported the presence of small scattered neuronal ganglia in the mucosal layer (mucosal ganglia) and between the longitudinal muscular layer and serosa (subserosal ganglia), as well as a ganglionated plexus within the muscularis mucosae [31,61].

Enteric neurons are characterized by a high degree of differentiation in terms of morphological, functional and electrophysiological properties [25]. Moreover, the enteric neurons are also highly diverse with regard to their ability to synthesize neuronal active substances. Apart from acetylcholine (the main neuromediator in the ENS), a wide range of other neuronal factors have been described in enteric nervous structures [25,62,63,64,65]. The most important neuronal factors include vasoactive intestinal polypeptide (VIP), substance P (SP), galanin (GAL), nitric oxide and calcitonin gene-related peptide (CGRP). These substances may act as neuromediators and/or neuromodulators and participate in many regulatory processes including, among others, intestinal motility, secretion in the GI tract, immunological processes, blood flow, sensory stimuli conduction, intestinal digestion and absorption [16,23,25,62,63,64,65]. It should be noted that several active substances have been noted in the enteric neurons. Their exact roles are often still not quite clear. It is also known that the roles of the particular neuronal factors in the regulation of the stomach and intestine activity may depend on the segment of the GI tract and animal species studied. Such a substance is GAL and its participation in the control of the intestinal motility. Previous studies have shown that GAL induces the contraction of the ileal smooth muscles in the rat, guinea-pig rabbit and pig [66], while in the canine ileum and stomach it shows relaxant effects [67]. A similar situation is observed in the case of SP, which strongly stimulates the contraction of intestinal muscles in the rat and dog, while in humans such activity is rather limited [68,69,70]. Moreover, one substance very often appears to be involved in various GI tract activities. For example, CGRP (which is known as a key factor in sensory and pain stimuli conduction within the GI tract [71,72]) may also participate in the regulation of intestinal motility, mesenteric and intramural blood flow, gastric secretion, absorption of the nutrients in the intestine and protective reactions [73,74,75,76,77]. A detailed discussion of the exact functions played by all neuronal factors located in the enteric neurons is almost impossible because new active substances and their roles in various species are still being discovered. However, the main functions connected with the GI tract of selected neuronal substances occurring in the ENS are presented in Table 1.

In addition to neurons, the ENS also includes numerous glial cells, which are called enteric glial cells (EGC) [141,142,143]. Glial cells in the gastrointestinal tract are generally characterized by small size, irregular or stellate shape and numerous processes which are in direct contact with neuronal cell bodies and nerve fibers. Based on previous studies, it is known that ECG may be divided into four major types, and classification of the EGC is similar to that used in the case of glial cells in the central nervous system [144,145]. The first type is “protoplasmic” glial cells (type I glial cells), which are located between neuronal cells in the enteric ganglia and their appearance resembles protoplasmic astrocytes in the brain. The second type of glial cells (type II glial cells) are “fibrous” glial cells, whose processes accompany the nerves connecting the enteric ganglia with each other. These cells are similar to fibrous glial cells located in the central nervous system. Moreover, mucosal glial cells (type III glial cells) located near nerve fibers in the mucosal layer and intramuscular glial cells (type IV glial cells) accompanying the nerve fibers in the muscular layer have been described in the gastrointestinal tract [144,145].

It was initially thought that glial cells are only structural support to neurons, but it is now known that EGC play multidirectional functions in the regulation of various aspects of the ENS and all gastrointestinal tract activities [146,147]. Primarily, they take part in processes connected with the development, protection and nutrition of the enteric neuronal cells [148,149]. They regulate growth, maturation and differentiation of the enteric neurons, and affect synthesis and release of neuromediators and/or neuromodulators, thus constituting a key factor in maintaining intraneuronal homeostasis [145,149,150,151].

Moreover, EGC (especially mucosal glia) are involved in activities of the intestinal barrier integrity and functions [152]. It is known that EGC synthesize a wide range of several substances, such as glial-derived neurotrophic factor, transforming growth factor-β1 and neurotrophins, and act on the intestinal epithelial cells through paracrine mechanism [152,153]. Experimental studies have also shown that in animals with genetic ablation of EGC, the intestinal epithelial layer loses its integrity and disturbances in vascularization appear and lead to severe inflammatory processes [154].

Enteric glial cells also have important functions during intestinal pathological states. They participate in immune cell modulation in a wide range of the intestinal diseases, including ulcerative colitis, Crohn’s disease and colorectal cancer [142]. During inflammatory processes, proliferation of EGC occurs [155]. Glial cells participate in the immune recognition of pathological stimuli and may act as antigen-presenting immune cells [156]. Moreover, an increase in the production of some cytokines, including, among others, interleukins (IL-1β and IL-6) [157,158,159], as well as nerve growth factor (NGF) [160], glial fibrillary acidic protein (GFAP) [161] and nitric oxide (NO) [162] in glial cells has been noted during inflammatory processes.

It is also known that enteric glial cells play important roles in the pathogenesis of neurodegenerative diseases, including Parkinson’s, Alzheimer’s and Creutzfeldt-Jakob diseases. They are considered to be a possible trigger point for neurodegenerative processes, which through the gut–brain axis may efficiently affect neurodegenerative processes in the central nervous system [143,144,145,146,147,148,149,150,151,152,153,154,155,156,157,158,159,160,161,162,163].

An important feature of enteric neurons is the ability to change their morphological, physiological and neurochemical properties under the impact of physiological and pathological factors [15,65,164]. Changes in the ENS have been described during growth and aging, diet changes, as well as various intestinal pathological processes, systemic diseases and the impact of toxic substances [15,65,164,165]. Changes in enteric neurons are a sign of the adaptive and protective reactions and contribute to homeostasis maintenance in the GI tract [164,165]. Moreover, such changes appearing under the impact of disease or toxic substances may be the first signs of subclinical pathological processes or toxicity [166]. Some studies have indicated that mycotoxins may affect the morphology and neurochemical character of the enteric neurons. The following is a short characterization of several mycotoxin-induced changes in the enteric nervous system.

## 3. Mycotoxins Affecting the Enteric Neurons

### 3.1. Deoxynivalenol

Deoxynivalenol (DON—molecular weight 296.31 g/mol), belongs to the trichothecene family and is a substance produced by Fusarium spp. [9]. It is commonly found in barley, oat, rye, corn and rice [167,168]. The signs of toxicity depend on the dose, mammal species and duration of exposure. The most frequent symptoms of toxicity with DON include loss of appetite, decreased body weight gain, neuroendocrine disorders, vomiting and diarrhea [169].

In the GI tract, toxicity with DON results in a wide range of histopathological changes, such as inflammatory infiltration, necrotic changes in the intestinal villi, edema of lamina propria, a decrease in the number of goblet cells in the jejunum and the ileum, intensification of apoptosis and degeneration of lymphoid cells in the GI tract [10,170]. These changes, together with DON-induced disturbances in the synthesis of many active substances produced by the gastrointestinal mucosa lead to the injury of the intestinal barrier and abnormal nutrient absorption [11].

Within the nervous system, DON-induced changes include abnormal synthesis of neuronal neurotransmitters and/or neuromodulators and in disturbances in neuronal activity [171,172]. Moreover, in neuronal cells, DON induces apoptosis, affects the cerebral lipid peroxidation and influences neuronal calcium homeostasis, and these disturbances in the neuronal cells may lead to anorexic actions [172,173].

During a study performed on male Wistar rats (*Rattus novergicus*) aged 21 days, the influence of relatively low doses of DON on the ENS was described [14]. In that experiment, DON in various doses (from 0.2 mg/kg of chow to 2 mg/kg of chow) was given for 42 days, and the ENS was studied using immunohistochemistry and microscopic analysis. It was shown that this mycotoxin does not affect the myenteric ganglia organization in the jejunum [14]. Between control animals and rats receiving DON there were no differences in the density of glial cells located in the myenteric plexus, or the population density of myenteric neurons. Moreover, DON did not change the density of particular subpopulations of the myenteric neurons, i.e., cholinergic and nitrergic neurons [14]. However, all concentrations of DON studied in the above-mentioned experiment caused a decrease in the area of the general population of the myenteric neuronal cells, as well as cholinergic and nitrergic cell neurons. Moreover, DON also decreased the area of gliocytes located in the myenteric plexus [14] and decreased the myenteric ganglia area. It should be noted that during the cited study, besides changes in the ENS, the animals did not show any other symptoms of toxicity, including a decrease in body weight, diarrhea, loss of appetite or changes in the oxidative status [14]. This indicates that changes in the ENS are the first symptoms of toxicity with low doses of DON.

### 3.2. T2 Toxin

T2 toxin (molecular weight 466.5 g/mol), similar to DON, belongs to the trichothecene family of toxins. It is mainly synthesized by *Fusarium sporotrichioides, F. langsethiae, F. acuminatum* and *F.*
*poae* and is recognized as the most acutely toxic trichothecene [174]. The impact of T2 toxin on the GI system manifests itself by (among others) histopathological changes in the intestinal mucosal layer (even with low doses), disturbances in the intestinal barrier functionality, influence on the enzymatic activity of enteric cells and inhibition of mucin production [175,176,177,178].

T2 toxin also shows neurotoxic activity and exposure to this substance results in a wide range of neurological symptoms, such as ataxia, muscular weakness, anorexia, as well as pathological lesions in the brain with disturbances in the functioning of this organ [179,180,181]. The main mechanisms underpinning the neurotoxic properties of T2 toxin are connected with reactive oxygen species and oxidative stress, as well as with mitochondrial dysfunction (consisting of the inhibition of the mitochondrial membrane potential and intensification of apoptosis) [182].

The ENS was studied using immunofluorescence in an experiment in vivo performed on juvenile (8-week-old) female domestic pigs of the White Large Polish Breed subjected to oral administration of T2 toxin at the level of 12 µg/kg body weight/day for 42 days [15]. Significant changes in the neurochemical character of the enteric neurons and nerve fibers located in the GI tract wall were described in this study. The character of changes depended on the type of the enteric plexus and the intestinal segment. It was reported that the administration of T-2 toxin increases the number of enteric neurons containing VIP in the porcine stomach and duodenum. These changes concern both myenteric and submucous plexuses and they are more visible in the duodenum, especially in the myenteric and outer submucous plexuses [15]. The same study showed that T-2 toxin also increases the number of nerve fibers containing VIP located in the muscular and mucosal layers of the porcine stomach and duodenum [15]. As previously indicated (Table 1), VIP in one of the potent inhibitory factors in the ENS and causes the hyperpolarization and relaxation of the gastrointestinal muscles and sphincters [132,133]. Moreover, VIP (as a vasodilator) increases blood flow in the wall of the GI tract and mesentery [132,134]. This substance may also affect the secretory activity of the GI tract, and the character of this activity depends on the GI tract segment [138,139,140]. It is known that VIP inhibits the gastric acid secretion in the stomach, but stimulates the secretion of the intestinal juice. VIP also has neuroprotective properties and increases the survivability of the enteric neurons [131]. Moreover, it is involved in immunological processes and shows anti-inflammatory properties. VIP also inhibits macrophages and inhibits the secretion of pro-inflammatory factors [135,136,137]. It is assumed that the increase in the number of VIP-positive enteric nervous structures under the impact of T2 toxin is connected with the protective and anti-inflammatory properties of VIP.

The influence of T2 toxin on the number of the enteric neurons containing cocaine and amphetamine-regulated transcript (CART) has been also reported [183]. In this study, T2 toxin was orally administrated to juvenile sows of the Large White Polish breed in a dose of 200 µg/kg of feed (the suggested permissible level of this toxin in the feed for pigs) for 42 days and the immunoreactivity in the ENS was evaluated using immunofluorescence. After the administration of T2 toxin, an increase in the number of CART-positive enteric neurons in all types of enteric plexuses as well as the number of nerve fibers containing CART in the mucosal and muscular layers in the stomach, duodenum and descending colon were described. The most visible changes were noted in the submucous plexus in the stomach and inner submucous plexus in the descending colon, where the number of CART-positive nerves under the impact of T2 toxin more than doubled [183]. It should be underlined that the exact functions of CART in the ENS are not clear [184]. A few studies concerning this issue have shown that CART inhibits the secretion of hydrochloric acid in the stomach and influences colonic motility [91,92]. This activity is probably done via the gut–brain axis because the direct impact of CART on isolated intestinal muscles does not cause changes in intestinal muscle contractility. The regulation of intestinal activity through the gut–brain axis is more likely since CART is known as an important factor regulating the feeding behavior in the central nervous system [185]. Moreover, numerous studies in which an increase in CART levels in the ENS has been observed strongly suggest that this peptide also takes part in protective and adaptive reactions in response to pathological, toxicological and physiological factors [46,166,184].

Another study (also performed with the immunofluorescence technique) concerning the impact of T2 toxin on the ENS was also performed on juvenile female pigs of the Large White Polish breed, which were treated with given T2 toxin orally in the dose of 12 µg/kg body weight/day for 42 days [16]. In this study, it was shown that T2 toxin affects the population of neurons containing calcitonin gene-related peptide located in the enteric plexuses in the porcine descending colon [16]. The administration of T2 toxin caused an increase in the number of CGRP-positive neurons in the myenteric, outer submucous and inner submucous plexuses, as well as an increase in the density of intramucosal nerves immunoreactive to these neuronal factors, without changes in the number of CGRP-positive nerve fibers in the muscular layer [16]. Moreover, it was shown that T2 toxin changes the neurochemical character of CGRP-positive neuronal cells, which were expressed by fluctuations in the degree of co-localization of CGRP with other neuronal factors (including substance P, nitric oxide synthase, galanin, CART peptide and vesicular acetylcholine transporter) in the same enteric nervous structures [16].

CGRP is a substance which primarily occurs in sensory neurons and is involved in sensory and pain stimuli conduction [71,72,93,94,95]. Moreover, CGRP in the GI tract takes part in the regulation of intestinal motility and increases blood flow in the mesenteric vessels [94,96,97,98,99]. It is also known that CGRP inhibits gastric acid secretion in the stomach with simultaneous induction of somatostatin release and regulates the absorption of nutrients from the intestine [186]. Previous studies have also shown that CGRP takes part in inflammatory processes in the intestine [99,100,101]. The multidirectional functions of CGRP in the ENS appear to be confirmed by a wide range of other neuronal substances present in CGRP-positive enteric neurons (such as substance P, nitric oxide CART peptide and galanin) which also play various roles in the GI tract (Table 1).

These reports of the influence of T2 toxin on the expression of a wide range of neuronal factors responsible for various regulatory processes in the ENS [15,16,183], strongly suggest that even relatively low doses of this mycotoxin may influence various intestinal activities, such as motility, secretion, conduction of sensory stimuli and regulation of the blood flow in the intestinal wall [15,16,183].

### 3.3. Zearalenon

Zearalenon (ZEN—molecular weight 318.364 g/mol) is synthesized mainly by *Fusarium graminearum, culmorum, crookwellense* and *roseum* and is found in barley, oat, wheat and bread [187]. The toxicity of ZEN is connected with its chemical structure, which allows it to act on the estrogen receptors, which are present in many internal organs [8]. ZEN can cross the blood-brain barrier and may influence neurons in the central nervous system [188,189]. It has been shown that exposure to ZEN leads to the abnormal synthesis of neuronal factors and enzymes in the brain neurons, induces apoptosis of the neuronal cells, increases oxidative stress reactions, influences the development of the nervous system, may cause behavioral aberrations and affects glial cell functions [189,190,191,192]. In turn, in the GI system, ZEN (among others) disturbs intestinal homeostasis, changes intestinal microbiome, causes inflammatory cell proliferation and inflammation in the intestinal mucosal layer [11,193,194,195].

Although the impact of ZEN on the GI tract is relatively well known, studies concerning the influence of this mycotoxin on intestinal innervation are limited to two studies performed on the pigs of the Large White Polish breed (approximately 8 weeks old), in which the nervous structure was evaluated with the immunofluorescence technique [19,21].

These studies have shown that the administration of relatively low doses of ZEN—10 μg/kg body weight/day [19] or 0.1 mg/kg of chow/day [21], administered for 42 days affect the neurochemical coding of nerve fibers in the mucosal and muscular layers of the ileum. For the intramuscular nerves, these changes involved an increase in the number of fibers immunoreactive to CART, substance P, nitric oxide synthase, VIP and pituitary adenylate cyclase-activating peptide and a decrease in the number of fibers containing galanin [19]. In the mucosal layer, ZEN not only caused an increase in the number of nerve fibers containing SP and/or VIP, but also changed the morphology of these nerves [21]. In animals treated with ZEN, nerves immunoreactive to SP and/or VIP become thicker and more visible than in the control animals [21]. It should be underlined that all the above-mentioned neuronal factors play important multidirectional roles in the regulation of the intestinal activity both in physiological conditions as well as during pathological processes, the most important of which are listed in Table 1.

The impact of ZEN on the ENS in the porcine descending colon has also been reported. A study concerning this issue was performed on juvenile (8-week-old) female pigs of the Large White Polish breed, which were treated with a dose of ZEN at the level of 6 µg/kg b.w./day given orally for 42 days [16]. In this study, the ENS evaluation was conducted with the immunofluorescence technique. The impact of ZEN was similar to the influence of T2 toxin. ZEN increased the number of neurons containing CGRP (whose functions in the ENS are described in the subchapter concerning T2-toxin and presented in Table 1) in all types of the enteric plexuses located in the descending colon [16]. Moreover, ZEN-induced changes in the neurochemical character of CGRP-positive enteric neurons were also reported [16]. These changes consisted of an increase in the degree of co-localization of CGRP with other neuronal factors (including substance P, galanin, CART and nitric oxide synthase, which was used as a marker of neuron synthesized nitric oxide) in neurons within all types of the enteric plexuses and intramural nerve fibers [16]. The functions of the above-mentioned neuronal active substances are presented in Table 1.

### 3.4. Patulin

Patulin (PAT-molecular weight 154.12 g/mol) is produced by various species belonging to *Penicillium, Aspergillus, Paecilomyces* and *Byssochlamys* [196,197] and is present in fruits (especially in apples) and vegetables [196,197]. Previous studies have shown that exposure to patulin causes damage to the intestinal barrier and inflammatory processes in the GI tract and influences the gut microbiota and the production of the mucus by enterocytes [198,199]. The neurotoxic activity of PAT is also known. It causes damage to the DNA in brain neuronal cells, mitochondrial and lysosomal dysfunction, a reduction of ATP levels and intensification of oxidative stress reactions [200,201].

The influence of patulin on the enteric neurons has been the subject of only one study. This study was performed on the cell culture of the enteric neurons prepared from 2–3-month-old C57B6/J OlaHsd mice and included various methods, such as growth and viability testing, a cytotoxicity test, evaluation of calcium signaling, measurement of glucose content, neurite outgrowth measurement and a reactive oxygen species (ROS) test [202]. The enteric neurons were treated with *P coprobium* extract, which decreased their viability with a half-maximal effective concentration (EC_50_) of 1 ng/µL This study also showed that patulin affects excitability and glucose consumption of the enteric neurons, which results in a patulin-induced reduction of ATP levels and glucose concentration in the enteric neurons. It has been also reported that patulin causes disorders in calcium signaling in the enteric neurons and affects neuronal morphology, which results in a reduction of neurite outgrowth and total neurite mass [202].

### 3.5. Fumonisins

Fumonisins are synthesized by *Fusarium proliferatum* and *Fusarium verticillioides* and characterized by a high degree of toxicity [203]. Numerous types of these mycotoxins have been described, but the most toxicologically important are fumonisin B1 (molecular weight 721.838 g/mol), fumonisin B2 (molecular weight 705.83 g/mol) and fumonisin B3 (molecular weight 705.8 g/mol), due to their high levels in cereal grains and crop products [12,204]. Among the numerous internal organs and systems which may be affected by fumonisins, the nervous system is one of the most susceptible to the adverse effects of these mycotoxins. It is known that fumonisins may enhance neurodegenerative reactions and impair the developmental processes in neurons located in the central nervous system, and some studies have reported connections between exposure to these mycotoxins and the risk of neurodegenerative diseases, such as multiple sclerosis, Alzheimer’s disease and Parkinson’s disease [205,206]. Exposure to fumonisins also results in changes in the GI tract, which manifest as disturbances in intestinal absorption, changes in the enterocytes and abnormalities in the intestinal immunological processes leading to increased susceptibility to infections [20].

However, knowledge of the influence of fumonisins on the ENS is extremely limited. One study concerning this issue was performed using the immunohistochemistry method on male Wistar rats (*Rattus novergicus*), which were 21 days old [12]. This study showed that a mixture of fumonisin B1 and B2 added to food in doses of 1 and 3 mg/kg of body weight (i.e., in doses which may be present in “natural” conditions in the food of humans and animals) given for 63 days does not affect the organization of the myenteric plexus in the rat jejunum [12]. Such doses of fumonisins do not result in changes in the general number of myenteric plexus and the number of myenteric neurons causing nitric oxide synthase, which is a marker of structures synthesizing nitric oxide. However, some changes in the myenteric neurons were observed under the impact of the mentioned doses of fumonisins. These changes consisted of a reduction in the size (without changes in their morphology) of neurons located in the myenteric plexus and included both neurons immunoreactive to pan-neuronal marker HuCD and nitric oxide synthase. Suoza et al. (2014) [12] reported that fumonisins not only affect the development and growth of neurons in the central nervous system but may also influence these processes in the ENS.

The influence of fumonisins on the ENS in the rat duodenum and jejunum of adolescent (5-weeks-old) male Wistar rats was also studied by Rudyk et al. (2020), using the immunohistochemistry method and histomorphometric analysis [13]. A mixture of fumonisins B1 and B2 were administered in a dose of 90 mg/kg of body weight for 21 days. That study demonstrated that fumonisins influence the following parameters within myenteric and submucous plexuses: area, perimeter, mean Feret diameter, mean diameter and sphericity [13]. It was also found that the impact of fumonisins on the ENS depends on the segment of the GI tract and the type of the enteric plexus. Fumonisin-induced changes in the duodenum were less visible, concerned only the submucous plexus and consisted of a reduction of area and mean diameter of ganglia, while the other parameters in the submucous plexus and all parameters studied in the myenteric plexus were not subjected to change. In the jejunum, changes were noted in the myenteric and submucous plexuses and consisted of an increase in the sphericity of ganglia and a reduction of other parameters in both types of plexuses. Moreover, the most visible changes were noted in the myenteric plexus.

The mechanisms of the impact of fumonisins on the ENS are unknown, but they probably inhibit ceramide synthase—an enzyme contributing to sphingolipid synthesis [207].

## 4. Mycotoxin Consumption and Human Gastrointestinal Diseases

The multidirectional adverse effects of mycotoxins on the GI tract (Table 2) cause that exposure to these substances may result in various disturbances of the GI activity in humans. However, the common prevalence of mycotoxins in the human environment and food indicates that participation of these chemicals in the development of intestinal diseases in humans may be an important public health problem all over the world [208].

The impact of mycotoxins on the intestinal barrier functions, intestinal immunity, secretory activity and gut microflora, as well as their genotoxic/mutagenic and carcinogenic effects are mainly known from experimental studies (Table 2). Such studies do not always fully reflect the conditions of natural exposure to mycotoxins. The first problem is the dose of mycotoxins, which is very difficult to determine in the human diet [254,255]. The second, more important, problem is the fact that food may contain several or even a dozen mycotoxins at the same time. These mycotoxins may chemically interact with each other, which leads to changes in their toxic properties and bio-availability. In this case, synergistic interactions between mycotoxins is particularly dangerous [255,256]. For example, previous studies have shown that mixtures of ZEN and DON or DON, T2 and ZEN show higher toxicity than these individual mycotoxins [175,257]. Moreover, it is known that human food may also contain other active substances and contaminations, such as bacterial products, pesticides, phytotoxins, chemical contaminations and preservatives, which not only affect mycotoxin activity but may contribute to various disorders in the GI tract [258]. That is why it is so difficult to determine the effective participation of mycotoxins in the development of human gastrointestinal diseases.

A comparison of histopathological changes occurring in the GI tract during human gastrointestinal diseases and changes in the intestine caused by mycotoxins has shown that the negative development in the GI tract in both cases are similar [255]. This may suggest a correlation between a degree of exposure to mycotoxins and the risk of human gastrointestinal diseases, as well as the participation of mycotoxins in the development of various diseases, including inflammatory bowel disease, Crohn’s disease, coeliac disease and colorectal cancer [255]. However, only comprehensive epidemiological studies on the relationships between mycotoxin levels in food, blood and urine and the occurrence of particular diseases conducted on a large human population would explain the connection between exposure to mycotoxins and the risk of human gastrointestinal diseases. Unfortunately, such studies are fragmentary and relatively few. These studies have reported that aflatoxins (especially aflatoxin B1) may pose a carcinogenic risk and exposure to these chemicals may increase the risk of gastric and colorectal cancer [259,260]. Other studies suggest a correlation between the exposure to ZEN and colorectal cancer [261], as well as relationships between exposure to aflatoxins and Crohn’s Disease, coeliac disease and ulcerative colitis [262]. Despite this, differences in concentration of patulin and citrinin in plasma and urine between healthy people and patients suffering from colorectal cancer have not been observed, which may suggest that these mycotoxins are not key factors leading to this disease [263].

## 5. Conclusions

Based on previous studies, it is known that mycotoxins affect the enteric nervous system (Table 3). This impact may be multidirectional and depends not only on the chemical structure of the mycotoxin and mammal species studied, but also on the type of the enteric plexuses and segment of the digestive tract. Mycotoxins may act on the size and morphological properties of intestinal nervous structures and the neurochemical character of the enteric neurons. These changes are probably a result of adaptive and protective reactions, which affect homeostasis maintenance. Moreover, mycotoxin-induced changes in the ENS are often the first sign of exposure to low doses of mycotoxins. Understanding the exact mechanisms connected with the influence of mycotoxins on the intestinal innervation may be very important in determining mycotoxin dose limits, which are safe and neutral for the living organism. Unfortunately, the current information about the influence of mycotoxins on the ENS is relatively limited and elucidation of all aspects connected with this issue requires further research.

## Figures and Tables

**Figure 1 toxins-12-00461-f001:**
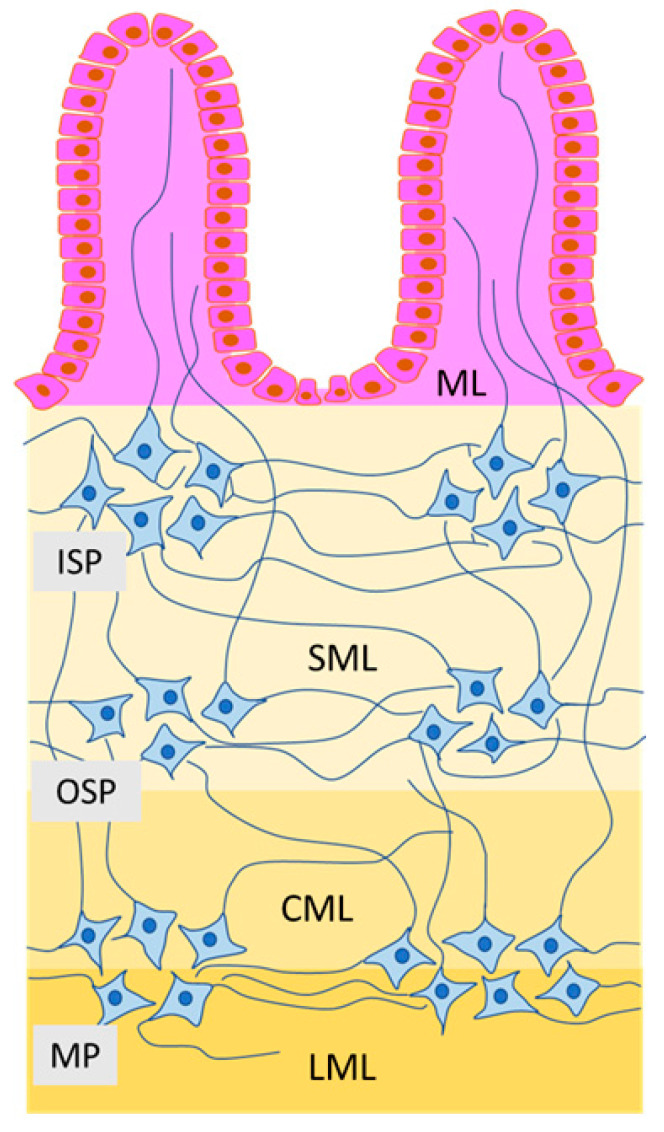
Organization of the enteric nervous system in the intestine of the domestic pig: MP—myenteric plexus, OSP—outer submucous plexus, ISP—inner submucous plexus, LML—longitudinal muscular layer, CML—circular muscular layer, SML—submucosal layer, ML—mucosal—layer.

**Table 1 toxins-12-00461-t001:** Functions of selected active substances in the enteric nervous system.

Active Neuronal Substance in the ENS (Alphabetical Order)	Selected Functions	References
Acetylcholine (Ach)	Stimulation of the intestinal motility	[78,79,80]
	Stimulation of electrolyte, water, enzymes and hormones secretion	[81,82,83,84]
	Participation in protective mechanisms	[82,85,86]
	Ant-inflammatory and immunostymulatory effects	[87,88,89]
	Blood flow regulation	[90]
Cocaine and Amphetamine Regulated Transcript (CART)	Inhibition of gastric acid secretion	[91]
	Regulation of the intestinal motility	[92]
Calcitonin Gene-Related Peptide (CGRP)	Participation in sensory and pain stimuli conduction	[71,72,93,94,95]
	Regulation of the intestinal motility	[94]
	Blood flow regulation	[96,97,98,99]
	Protective roles	[73,99,100,101]
	The influence on intestinal absorption	[74]
Galanin (GAL)	Intestinal motility regulation	[66,67,68,69,70,102]
	Influence on secretory activity	[103,104,105]
	Participation in inflammatory processes	[103,105,106]
Nitric Oxide (NO)	Inhibition of the intestinal motility	[106,107,108]
	Participation in inflammatory processes	[109]
	Regulation of intestinal secretion, water and electrolyte transport	[110,111,112,113]
	Regulation of blood flow	[114,115]
	Participation in inflammatory processes	[116,117]
Pituitary Adenylate Cyclase-Activating Polypeptide (PACAP)	Inhibition of the intestinal motility	[118,119]
	Stimulation of gastric secretory activity	[120,121]
	Regulation of ion transport and Luminal fluid regulation in the large intestine	[121,122,123]
	Regulation of blood flow	[124]
Substance P (SP)	Protective roles	[100]
	Sensory stimuli conduction	[93,125]
	Regulation of the intestinal motility	[125,126,127]
	Regulation of water and electrolytes secretion	[125,128,129]
	Participation in inflammatory processes	[125,130]
Vasoactive Intestinal Polypeptide (VIP)	Neuroprotective functions	[131]
	Regulation of the intestinal motility	[132,133]
	Vasodialtory activity	[132,134]
	Participation in intestinal immunomodulation	[135,136,137]
	Influences on intestinal secretion	[138,139,140]

**Table 2 toxins-12-00461-t002:** Gastrointestinal signs and effects of mycotoxins on the gastrointestinal tract.

Mycotoxin	Gastrointestinal Signs of Toxicity	References	Influence on the Digestive Tract	References
Doxynivalenol (DON)	Abdominal pain, increased salivation, diarrhea, vomiting, anorexia, decrease body weight gain	[169,209,210,211,212,213,214]	IPEC-J2 cell line from porcine jejunal epithelium: cytotoxicity, decrease in transepithelial electrical resistance, disruption of epithelial integrity	[176]
			Porcine jejunal explant samples: shortened and coalescent villi, lysis of enterocytes, edema, upregulation of proinflammatory cytokines expression	[215,216]
			Pigs of White Large Polish Breed: increase in the mucosal thickness and the intestinal crypt depth, atrophy of the villi, changes in the number of goblet cells, inflammatory infiltration, intensification of apoptosis, changes in ultrastructure of intestinal cells	[10,11,175,214,217,218]
			Human Colonic Cell Lines Caco-2, T84, HT-29: decrease in cell proliferation, changes in permeability, genotoxicity, intensification of apoptosis, increase in the expression of proinflammatory cytokines, influence on DNA synthesis	[215,219,220,221]
			Poultry: decrease in the high of villi	[222,223]
T2 Toxin	Gastrointestinal bleeding, diarrhea, vomiting, decreased feed consumption and weight gain	[224,225,226]	IPEC-J2 cell line from porcine jejunal epithelium: cytotoxic effects, disruption of intestinal barrier integrity	[176]
			human intestinal Caco-2 cells disturbances in intestinal barrier, enzymatic activity of enteric cells, inhibition of mucin production	[178]
			Pigs of White Large Polish Breed or crossbred pigs: congestion and hemorrhage of the gastrointestinal mucosal layer, inflammatory infiltration, in high doses—necrotic changes	[175,227,228,229]
			Sprague-Daw-ley rats: inflammatory and necrotic changes in, lymphocytic necrosis in intestinal Peyer’s patches, influence on nutrients absorption, influence on DNA synthesis	[230,231,232]
Zearalenone (ZEN)	Gastrointestinal symptoms are not typical for ZEN toxicity. Decrease in feed intake and body weight, changes in intestinal microbiome	[195,233]	Pigs of various breeds: increase in the mucosal thickness, increase in the number of goblet cells, increase in lymphocyte number in epithelium, intensification of apoptosis, influence on enzymatic activity of mucosal cells, changes in intestinal microbiome	[10,11,175,193,194,195,234,235]
			Intestinal porcine epithelial cell line (IPEC-1): influence on cell activity by changes in gene expression	[236]
			Poultry: changes in the high of intestinal villi	[237]
Patulin (PAT)	Anorexia, salivation, distended abdomen loss of body weight, bleeding from the digestive tract and diarrhea	[238,239,240,241,242,243]	Human intestinal Caco-2 cells: the influence on permeability and ion transport in the mucosa, epithelial desquamation and sub mucosal swelling, genotoxicity effects, modulation of tight junctions	[198,199,244]
			Rodents: mucosal layer injury, ulceration, fibrosis in the sub mucosa, necrosis	[238,239,240,241,242]
			Porcine jejunal explant samples: villi atrophy and necrosis, decrease in the number of goblet cells, increase in cell apoptosis	[245]
Fumonisins (FUM)	reduction of feed consumption and body weight, abdominal pain, diarrhea	[246,247,248,249]	Human Colonic Cell Lines Caco-2, HT-29: growth inhibition and apoptosis induction, impact on mitochondrial metabolism, necrosis	[221,250]
			Rodents: inflammatory infiltration increase in the number of mitotic figures in the intestinal crypts, necrotic changes	[251,252]
			Intestinal porcine epithelial cell line (IPEC-1): inhibition of cell proliferation, intestinal barrier dysfunction	[253]

**Table 3 toxins-12-00461-t003:** Influence of mycotoxins on the enteric nervous system.

Mycotoxin	Dose Examined	Animal Species or Kind of Tissues	Experimental Method Used in the Study	Character of Changes in the ENS	References
Doxynivalenol	from 0.2 mg/kg of chow to 2 mg/kg of chow	Wistar rats (*Rattus novergicus*)	immunohistochemistry and microscopic analysis	Reduction of the area of general population of the myenteric neurons, glial cells in the myenteric plexus and whole myenteric ganglia.	[14]
T2 Toxin	12 µg/kg body weight/day	domestic pig of the White Large Polish Breed	Immunofluorescence method and microscopic analysis	Increase in the number of VIP-positive enteric neurons and intramucosal and intramuscular nerve fibers containing VIP in the stomach and duodenum.	[15]
	200 µg/kg of feed	domestic pig of the White Large Polish Breed	Immunofluorescence method and microscopic analysis	Increase in the number of CART-positive enteric neurons and intramucosal and intramuscular nerve fibers containing CART in the stomach, duodenum and descending colon.	[55]
	12 µg/kg body weight/day	domestic pig of the White Large Polish Breed	Immunofluorescence method and microscopic analysis	Increase in the number and changes in neurochemical character of CGRP-positive enteric neurons in the descending colon.	[16]
Zearalenon	10 μg/kg body weight/day	domestic pig of the White Large Polish Breed	Immunofluorescence method and microscopic analysis	Increase in the number of nerve fibers immunoreactive to CART, SP, NOS, VIP, PACAP and decrease in the number of GAL-positive nerve fibers in the muscular layer of the ileum.	[19]
	0.1 mg/kg of chow/day	domestic pig of the White Large Polish Breed	Immunofluorescence method and microscopic analysis	Increase in the number of nerve fibers immunoreactive to SP and VIP with changes in their morphology	[21]
	12 µg/kg body weight/day	domestic pig of the White Large Polish Breed	Immunofluorescence method and microscopic analysis	Increase in the number and changes in neurochemical character of neurons immunoreactive to CGRP in the descending colon.	[16]
Patulin	EC50 = 1 ng/µL	culture of the enteric neurons from C57B6/J OlaHsd mice	Growth and viability testing, cytotoxicity test, evaluation of calcium signaling, measurement of glucose content, neurite outgrowth measurement and reactive oxygen species (ROS) test	Reduction of ATP levels and glucose concentration, disorders in calcium signaling in the enteric neurons, changes in their morphology.	[71]
Fumonisins	1 and 3 mg/kg body weight	Wistar rats (*Rattus novergicus*)	immunohistochemistry method	Reduction of the size of neurons in the enteric ganglia.	[12]
	90 mg/kg body weight	Wistar rats (*Rattus novergicus*)	immunohistochemistry method and histomorphometrical analysis	Reduction of area and mean diameter of the submucous plexuses in duodenum. Reduction of area and mean diameter of myenteric and submucous plexuses in the jejunum, increase of sphericity of the enteric ganglia.	[13]

VIP—vasoactive intestinal polypeptide; CART—cocaine- and amphetamine-regulated transcript; CGRP—calcitonin gene related peptide; SP—substance P; NOS—nitric oxide synthase; PACAP—pituitary adenylate cyclase activating peptide; GAL—galanin.

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
