# Peer review of "Mycotoxins and the Enteric Nervous System"

_toxins, 2020, doi:10.3390/toxins12070461_

Round 1
Reviewer 1 Report
The authors present a review of the literature to outline the effects of some mycotoxins of food origin on the enteric nervous system.
Although the subject is very interesting, English is very poor, so many paragraphs are almost incomprehensible. The sentences sound intricate and too long. Many paragraphs are confusingly written. Sometimes references are made to humans, sometimes to animal models with no explanation in the text. There are also real mistakes when skipping from one model to another.
Major Revisions
1-Do a full English revision by an expert
2- lines 63-66 The authors indicate the presence of three plexuses as constituents of the enteric nervous system. In order to write a review that has enteric nervous system as its subject, one would need to know it very well. In humans there are only two plexuses. Their localization is also wrongly described. The submucosal plexus is located inside the submucosa. The myenteric plexus is located between the internal circular musculature and the external longitudinal musculature. This part must be rewritten. If, on the other hand, authors are not referring to humans, but to some animal, such as a pig, then the text must be rewritten because it is not clear. Reference 15 refers to porcine ENS.
3-Lines 76-78 The description of glial cell function is very poor. The role of glial cells is more extensive. I suggest to read a review on this topic and revise the part of manuscript that address on the role of these cells. Even the parallelism with astrocytes is not completely correct. In the CNS, astrocytes play roles that in the GI are exerted by the enteric glial cells in cooperation with immune cells.
4-Line 114-116 When you refer to an animal model you cannot simply call it a rat, mouse or pig. You have to include the genotype. You should also briefly summarize the experiment. For example, how were analyzed the ganglia mentioned in this paragraph? This should be applied throughout the text.
5- 141-151 / 211-217 / 229-231 / 232-235 etc.. The experimental models are never mentioned. This generates confusion and is not correct.
6- 152-170 Neuronal markers are never described or mentioned. Then, “XX”- positive neurons appear in the text (i.e, CART positive neurons, GCRP positive neurons, etc...) without explanation on their importance and functions.
7- 191-202 The effects of ZEN on neuronal markers are described in this section. An initial paragraph in which the enteric nervous system should be treated according to its chemical code is needed. Otherwise either the reader is already familiar with neurogastroenterology, or it is not possible to understand the message of the text.
8- Also in table 1 The experimental models are never mentioned. This generates confusion and is not correct
Author Response
The authors thank the Reviewer for insightful review, which allows to improve the manuscript.
1-Do a full English revision by an expert
The manuscript has been reedited and improved by native speaker in English
2- lines 63-66 The authors indicate the presence of three plexuses as constituents of the enteric nervous system. In order to write a review that has enteric nervous system as its subject, one would need to know it very well. In humans there are only two plexuses. Their localization is also wrongly described. The submucosal plexus is located inside the submucosa. The myenteric plexus is located between the internal circular musculature and the external longitudinal musculature. This part must be rewritten. If, on the other hand, authors are not referring to humans, but to some animal, such as a pig, then the text must be rewritten because it is not clear. Reference 15 refers to porcine ENS.
The fragment concerning the organization of the ENS has been extensively reedited (lines 55-108). It should be pointed out that concise description of the ENS in the first version of the manuscript did not result from poor knowledge of the ENS (some authors deal with the ENS from about 20 years) but from initial intention of the manuscript, which was to be maximally concise. However, the authors are in agreement with the Reviewer that the description of the ENS in old version of the manuscript was not clear. In new version the organization of the ENS has been described more precisely. Of course this description does not show all aspects connected with organization and functions of the ENS. As the Reviewer knows, the organization of the ENS may be the subject of long books and exact description all aspects of the anatomy and physiology of the enteric neurons exceeds the volume of one manuscript. However the authors do not agree with the sweeping statement that the human ENS in the intestine consists of only two kinds of plexuses. Of course the authors know the report of Graham et al (Gastroenterology. 2020, 158, 2221-2235.e5), who write that submucosal plexuses in human colon are disseminated and do not form separate plexuses, but the majority of previous studies have shown that there are at last three types of the enteric plexuses in the human intestine. The authors of the manuscript during studies on human intestine also usually noted three plexuses: myenteric and two submucosal plexuses (the authors studied the ENS on the sections not whole mounts and of course the way of preparation of tissues may be important in this case). Without getting into a further polemic, in the opinion of the authors in the review report the knowledge should be presented in impartial manner and therefore in new version of the manuscript various views on organization of the ENS in human intestine has been presented (lines 83-97).
3-Lines 76-78 The description of glial cell function is very poor. The role of glial cells is more extensive. I suggest to read a review on this topic and revise the part of manuscript that address on the role of these cells. Even the parallelism with astrocytes is not completely correct. In the CNS, astrocytes play roles that in the GI are exerted by the enteric glial cells in cooperation with immune cells.
The glial cells in the ENS are not the main subject of the manuscript. However, the authors are in agreement with the Reviewer that old version of the manuscript was not clear in this regard. Therefore, the description of enteric glial cells has been reedited and completed (lines 137-176).
4-Line 114-116 When you refer to an animal model you cannot simply call it a rat, mouse or pig. You have to include the genotype. You should also briefly summarize the experiment. For example, how were analyzed the ganglia mentioned in this paragraph? This should be applied throughout the text.
According to Reviewer’s suggestion information about cited studies concerning the influence of mycotoxins on the ENS has been supplemented by available data (lines 206-209). On the other hand genotype of animals used in the previous studies performed by other authors are often not mentioned within the reports. Moreover doses of mycotoxins, and methods used in the cited study has been added. On the other hand the more precisely description of cited studies is not usually needed in typical review reports, because references allow the reader to find and read the whole cited study
5- 141-151 / 211-217 / 229-231 / 232-235 etc.. The experimental models are never mentioned. This generates confusion and is not correct.
In all these cases the additional information has been added (lines 235-239, 258-261, 315-320,331-334, 353-359, 379-381, 393-395)
6- 152-170 Neuronal markers are never described or mentioned. Then, “XX”- positive neurons appear in the text (i.e, CART positive neurons, GCRP positive neurons, etc...) without explanation on their importance and functions.
According to Reviewer’s suggestion neuronal markers and their main functions have been presented in table 1 added into the manuscript. Moreover the main functions of neuronal active substances have been also added in lines 245-256, 266-275, 289-297)
7- 191-202 The effects of ZEN on neuronal markers are described in this section. An initial paragraph in which the enteric nervous system should be treated according to its chemical code is needed. Otherwise either the reader is already familiar with neurogastroenterology, or it is not possible to understand the message of the text.
The answer in point 6
8- Also in table 1 The experimental models are never mentioned. This generates confusion and is not correct
In table 1 (in new version of the manuscript table 3) – lines 469-474 short information about experimental models has been added
The authors hope that improvements of the manuscript and explanations will satisfy the Reviewer and will allow to publish the manuscript in Toxins

Reviewer 2 Report
Overall I think this is an informative mini review. I tried to improve the grammar, which the authors can view with my attached marked manuscript.
There were several instances of "neurochemical characterization" which I believe should be "neurochemical character".
Line 28: However, many aspects of the effects of mycotoxin activity on eukaryotic organisms are unknown.
Line 58: fragment, I'm not sure if this an anatomically correct word. I suggest segment or segmentation rather than fragment. Please check the entire document for the word fragment.
Line 86: intoxication- when I think of intoxication, it means drinking too much beer or wine! I suggest using toxicity. Please check the entire document for the word intoxication.
Line 260- Conclusion rather than summarization
Line 270- determination of mycotoxin dose limits which are

Author Response
The authors thank the Reviewer very much for positive revision of the manuscript
Overall I think this is an informative mini review. I tried to improve the grammar, which the authors can view with my attached marked manuscript.
The authors thank for grammar revision of the manuscript. All suggestions have been taken into account. Moreover, the manuscript has been reedited by native speaker in English
There were several instances of "neurochemical characterization" which I believe should be "neurochemical character".
The phrase “neurochemical characterization” has been replaced by the phrase “neurochemical character” in the whole manuscript.
Line 28: However, many aspects of the effects of mycotoxin activity on eukaryotic organisms are unknown.
The sentence has been corrected according to Reviewer’s suggestion
Line 58: fragment, I'm not sure if this an anatomically correct word. I suggest segment or segmentation rather than fragment. Please check the entire document for the word fragment.
The authors are in agreement with the Reviewer. the word “fragment” has been replaced by the word “segment” in the whole manuscript.
Line 86: intoxication- when I think of intoxication, it means drinking too much beer or wine! I suggest using toxicity. Please check the entire document for the word intoxication.
The authors are in agreement with Reviewer’s suggestion. “Intoxication” has been replaced by “toxicity”
Line 260- Conclusion rather than summarization
„summarization” has been replaced by “conclusion”.
Line 270- determination of mycotoxin dose limits which are
The sentence has been corrected according to Reviewer’s suggestion.

Reviewer 3 Report
Add two more tables, enrich the text and increase references to at least 150.
Author Response
Add two more tables, enrich the text and increase references to at least 150.
According to the Reviewer’s suggestions two new tables (tables 1 and 2 in new version) have been added into the manuscript. Table 3 (in old version table 1) has been enriched with additional data. The manuscript has been extensively reedited and enriched. The number of references has been enriched to 264. The manuscript has been corrected by native speaker in English

Reviewer 4 Report
In this work the Authors describe how mycotoxins influence the gut, especially the enteric nervous system (ENS). After a short introduction to ENS morphology and physiology the Authors detail the effects of Doxynivalenol, T2 toxin, Zearalenon, Patulin and Fumonisins on glial and neuronal cell populations, and how these toxins affect generally the GI tract.
Major remarks:
- The first figure is not informative in this form, it should have more detail, with more visual representation of toxin-related effects.
- The second figure/table, similarly should be reorganized for better understanding and more detail. Clinical symptoms and morphological changes in the ENS should be listed separately.
- Is there any literature reporting about the connection of chronic mycotoxin consumption and human diseases (coeliakia, IBD, GI cancers, etc?) If there is, it is an important translational aspect, that should be mentioned in the review.
- The manuscript needs thorough english language editing. Spelling is OK, but the composition of the text and the structure of sentences needs a lot of improvement.
Minor remarks:
- The molecular weight of toxins should be mentioned in the text/2nd figure/table
- Instead the word "fragment" use "segment" or "part" regarding the GI tract
Author Response
The authors thank for review, which allows to improve the manuscript
Major remarks:
The first figure is not informative in this form, it should have more detail, with more visual representation of toxin-related effects.
Figure 1 was tasked with showing an organization of the enteric nervous system. the authors are in agreement that it was not informative as regards the influence of mycotoxins on the enteric neurons. In new version it has been reedited to show only an organization of the enteric nervous system. Regarding the scheme, which will show the influence of mycotoxins in the enteric neurons, the authors admit that they have no good idea. The influence of mycotoxins on the enteric nervous systems is multidirectional. Particular mycotoxins affect the enteric neurons in different ways. Moreover, mechanisms of such influence is not quite clear, and the knowledge on this issue is fragmentary. therefore in the opinion of the authors the best and the most readable way to show the impact of mycotoxins on the enteric nervous system is the table. Therefore the authors resigned from figures showing the influence of mycotoxins on the enteric nervous system.
The second figure/table, similarly should be reorganized for better understanding and more detail. Clinical symptoms and morphological changes in the ENS should be listed separately.
Table 1 (in new version table 3 – line 469) has been extensively reedited. Now it is more precise. The authors hope that new version of table is more understandable for readers. Moreover, new table (table 2) has been added – line 416. According to Reviewer’s suggestion this table shows mycotoxins-induced gastrointestinal clinical symptoms and influence of mycotoxins on the GI tract structure and functions.
Is there any literature reporting about the connection of chronic mycotoxin consumption and human diseases (coeliakia, IBD, GI cancers, etc?) If there is, it is an important translational aspect, that should be mentioned in the review.
According to Reviewer’s suggestion the subchapter about the connection between exposure to mycotoxins and risk of the gastrointestinal diseases in humans has been added (lines 409-450).
The manuscript needs thorough english language editing. Spelling is OK, but the composition of the text and the structure of sentences needs a lot of improvement.
The manuscript has been corrected by native speaker in English.
Minor remarks:
The molecular weight of toxins should be mentioned in the text/2nd figure/table
Molecular weights of mycotoxins have been added into the text of manuscript.
Instead the word "fragment" use "segment" or "part" regarding the GI tract
The word “fragment” has been replaced by the word “segment” in the whole manuscript
The authors hope that improvements will allow to publish the manuscript in Toxins.

Round 2
Reviewer 1 Report
The requested changes have been made
Now the English language is correct
Author Response
The authors thank for the review.
Reviewer 3 Report
The publication has been significantly improved, with new data and several new bibliographic sources.
Author Response
The authors thank for the review.
Reviewer 4 Report
Though still not perfect, English has improved significantly.
The tables are much more informative and include useful and valuable data for knowledge mining in the future.
Minor Remarks:
- Please edit the following table titles accordingly:
Table 2. Selected of gastrointestinal signs and effects of mycotoxins on the gastrointestinal tract
Table 3. Summarization of current knowledge about the influence of mycotoxins on the enteric nervous system
Author Response
All suggestions of the Reviewer concerning table titles have been taken into account.
Regards English language, as mentioned in the first answer, the manuscript has been corrected by native speaker in English, specialist in fields of biological and medical sciences, the worker of professional translation agency. University of Warmia and Mazury has signed an agreement with this agency. If necessary we can send certificate that the manuscript has been corrected. Please indicate what fragments of the manuscript need further language improvements. Then we will contact with the translation agency to further corrections.